# An Approach for Predictive Maintenance Decisions for Components of an Industrial Multistage Machine That Fail before Their MTTF: A Case Study

Francisco Javier Álvarez García [1,*] and David Rodríguez Salgado [2]

1 Department of Mechanical, Energy and Materials Engineering, University of Extremadura, C/Sta. Teresa de Jornet 38, 06800 Mérida, Spain
2 Department of Mechanical, Energy and Materials Engineering, University of Extremadura, Avda. Elvas s/n, 06006 Badajoz, Spain
* Correspondence: fjag@unex.es

**Abstract:** Making the correct maintenance strategy decision for industrial multistage machines (MSTM) is a constant challenge for industrial manufacturers. Preventive maintenance strategies are the most popular and provide interesting results but cannot prevent unexpected failures and consequences, such as time lost production (TLP). In these cases, a predictive maintenance strategy should be used to maintain the appropriate level of operation time. This research aims to present a model to identify the component that failed before its mean time to failure (MTTF) and, depending on whether the cause of the failure is known, propose the use of a predictive maintenance strategy and further decision-making to ensure the highest possible value from operating time. Also, it is necessary to check the reliable value of MTTF before taking certain decisions. For this research, a real case study of a MSTM was characterized component by component, setting the individual maintenance times. The initial maintenance strategy used for all the components is the preventive programming maintenance (PPM). If a component presents an unexpected failure, a method is proposed to decide whether the maintenance strategy should be changed, adding a predictive maintenance strategy to monitor said component. The research also provides a trust level to evaluate the reliable value of MTTF of each component. The authors consider this approach very useful for machine manufacturers and end users.

**Keywords:** predictive maintenance; multistage machine; sensorisation; decision-making; mean time to failure; algorithm; system

## 1. Introduction

Multistage machines (MSTM) are quite common in the manufacturing processes industry. These machines are more complex than single-stage machines. The diversity of components and coordinated steps or successive transformations they perform entails the need to establish an adequate maintenance strategy for each component.

It is very important to bear in mind that a failure in one of the components of a multistage industrial machine can lead to a failure in the whole machine. Due to this condition, the best maintenance policy must combine the most suitable strategies for each component. Different components of the same multistage machine may well have different maintenance strategies depending on their maintenance parameters that affect their mean time to repair (MTTR). Once the component has been repaired or substituted, the machine must return to its normal work rhythm and needs time to restart the line (TTLR).

The success in the use of these machines is to meet high demands without unexpected failures that involve the loss of production in progress and a high operation recovery time. Due to this, it is very important that the components of the machine are reliable. If these components have an individual MTTF, this time must be reliable to establish maintenance

policies that allow for optimizing the stop-operation time; it is necessary to have the right components, and reliable MTTF. Also it is very important that the main devices and location components used in the MSTM are correct, in order to eliminate avoidable failures. If the MTTF is reliable, it is therefore possible to program the preventive maintenance, so it is necessary to establish the adequate time to maintenance that does not affect the scheduled production.

Preventive maintenance is the most popular strategy in industrial manufacturing systems. Therefore, there must be an adequate level of stocks of components based on a mathematical model proposed in the decision-making strategy. The optimal decision process for setting time to production and time to maintenance programming is studied by A. Gharbi [1], namely, how to develop a mathematical model based on the cost for optimal decision making. A. Gharbi's [2] research also found the most appropriate production rate and preventive maintenance schedule that minimizes the total cost of maintenance and inventory/backlog in periodic preventive maintenance.

As already known, for established scheduled preventive times, is important to define what to do in these times. MTTR considers time to provisioning, time to replacement or removal of component, time to configuration or setting and time to mechanical adjustments. H. Jun-Hee [3] proposed in his research that periodic machine maintenance for single machines and flow-shop scheduling models should be based on an algorithm, minimizing the total weighted completion time. His work defines two principal maintenance actions, setup operations and removal operations, in a production system based on a sequence of single-stage machines. If a removal or setting time is required, a lateness time must be considered. As an in-line process needs to be functioning with stage coordination, is very important to measure the operation times and make the maintenance decisions. If the functioning of the MSTM requires setting maintenance operations during the times of normal operations, this can affect the cycle operation time of the whole machine, and some lateness times must be studied before the maintenance of the machine begins. Other studies have proposed how to realize the appropriate preventive maintenance with imperfect actions, while continuing the normal operation condition, as J. Zuhua [4] showed how to create function blocs in a Programming Logic Controller (PLC) with a previous data acquisition system.

But the important question for preventive maintenance is how to accomplish it, and what might be the appropriate procedure, depending on the system's definition and its complexity. Hernández, D.R. [5] modeled a discrete-time infinite horizon Markov Decision Problem, and F. Chiacchio [6] a stochastic-hybrid reliability model. Other studies, such as M. Fujishima [7], have calculated the optimal time to start preventive maintenance before an unexpected failure. A recent study by A. Irfan [8] modelled a series-parallel system, proposing a reliability model using a Lagrangian optimization method to guarantee MTTF values and avoid unexpected failures. Also, when an unexpected failure occurs, some essential information should be known. For example, the cause of the failure is important, whether the cause stems from a poor design of the machine or an incorrect location of the component, or whether the cause can be eliminated altogether to restore the machine's functioning with a higher level of availability and security. Also, it is important to determine whether the cause is a normal or an occasional (infrequent) situation.

All the components of a machine are always subject, at least, to the laws of degradation. Therefore, even working in its ideal operating conditions, the component will end up failing. In this sense, it would be appropriate to be able to calculate the reliability, as in D.M. Frangopol [9], of the component in the whole machine. However, this study is very complex and normally the manufacturer of the components only defines normal working conditions and sometimes the operating time. Therefore, it is necessary to study models that evaluate whether the component is suitable for the machine and if it is, whether it is so in the normal operation of the machine. G. Silva [10] proposes a model to decide on the most suitable maintenance strategy for the obsolescence of electronic components by creating a decision-making tool, and analyzing the risks, the obsolescence of the components and

the consequences of a failure. Recent research by Garcia, F.J.Á. and Salgado, D.R. [11] has proposed a matrix to decide the optimal preventive maintenance strategy based on the individual maintenance times of all the components, their approximate location (global operation condition (GOC)) in the machine, and two key performance indicators (KPIs). The results of both papers describe situations where the component may have different maintenance strategies. If a component fails multiple times, a failure mode and effect analysis (FMEA) can be the solution for finding a design error in the machine, in the component, or an inadequate component selection for the normal operation condition required by the machine. In this way, T. Yuk-Ming [12] has highlighted the importance of product design in the future reliability of the components that must work within certain operating conditions. Product design and functional performance have been shown to be the main research foci in this area.

Predictive maintenance strategies have been shown to be able to avoid unexpected failures by monitoring the operation of the machine using sensors (P. Ponce [13]) and machine learning algorithms to know the normal behavior of the components or of the machine. Dolatabadi, S.H. [14] has provided an overview of past articles highlighting the major expectations, requirements, and challenges for small and medium-sized enterprises (SMEs) regarding the implementation of predictive maintenance (PdM). Normally, the PdM based on algorithms have several steps: data acquisition, data manipulation, configuration, aggregation, and prediction model (the condition monitoring sub-model); and maintenance decision-making, scheduling, and status (the maintenance sub-model). Sometimes, the main algorithm or calculating process is embedded in a PLC, as discussed in Cavalieri, S. [15] and Bouabdallaoui, Y.S. [16].

The study by Garcia, F.J.Á. and Salgado, D.R. [17] described a way to present the available strategies for multistage industrial machines. Their paper describes preventive strategies (with or without stock) and developed predictive strategies like digital behavior twin (DBT), composed of an algorithm with no need to learn normal behavior. S. Givnan [18] studied the normal behavior of the components of an industrial machine for early failure detection by using a machine learning model based on feed-forward neuronal networks trained to identify normal and abnormal behavior. One of the best advantages of the algorithms and the machine learning models is the time necessary to train the model to identify the normal behavior of the machine. Industries need, to the degree possible, simple, fast and reliable systems to take decisions about the availability of their machines in order to avoid unexpected failures. M.M.L. Pfaff [19] developed and tested an adaptive algorithm in a real environment. This algorithm created a dynamic limit value using an adaptive characteristic value segmentation. The paper also studied the location of the sensors for predictive maintenance and confirmed that location can significantly affect the measurement result and, thus, has a direct impact on the outcome of the data analysis. One of the advantages of this research is that there is no need to train the algorithm; the application does not require in-depth process knowledge.

As the technical decisions to take maintenance actions can be provided by the analysis of technical data, normal behavior trained, or not trained, by the adopted predictive algorithm, some authors have mixed the machine learning study with the cost of the maintenance to take global predictive maintenance decisions, as in E. Florian [20] and S.Arena [21], by using, in this case, the Decision Tree technique (DTs) process of implementing predictive maintenance (PdM) and also detecting potential failures (identified through FMEA analysis) and evaluating direct and indirect maintenance costs. It is very important to evaluate a FMEA analysis where a possible failure design of the machine can be the reason of repeated failures.

The digital twin (DT) concept, based on cyber physical systems (CBS) (C. Stary [22]), is a good way to study predictive maintenance and the behavior of the machine if it works under different operating conditions. J. O'Sullivan [23] studied the adoption of digital twins by the maintenance engineering industry to aid in predicting problems before they occur. The algorithm used provided three alarm levels to identify action before a failure.

But not all MSTM can be modelled with a digital twin, due to the fact that these machines normally are highly customized and adapted to the needs of each end user, and therefore since they are not mass-produced, they would require the development of their specific digital twin.

New models embedded in industry 4.0 and created to control corrective maintenance actions are based on a system built on the augmented reality (AR) or computer vision (CV). These systems are used when the machine must be maintained with non-expert operators, and the support of the system can drive the maintenance action with the most success, and in the optimal time. As it can understand the machine, this supporting system minimizes the MTTR and lets the availability of the machine remain in the highest degree possible. Similarly, the work of Konstantinidis, F.K. [24] and Z. Haihua [25] also attempts to solve unexpected failures that are not stored in the maintenance-experience database.

Little of the literature focuses on the simultaneous study of different preventive and predictive maintenance strategies at the same time in the same system. In the case of the MSTM, such studies are non-existent. An interesting paper of H. Wang [26] focuses on a DT-enabled integrated optimization problem of flexible job shop scheduling and flexible preventive maintenance (PM), considering both machine and worker resources. This approach is interesting, particularly if it is possible to open a flexible window to preventive maintenance actions and let the system constantly work with the monitoring of predictive maintenance policy. The architecture of a DT-enhanced job shop is developed, and then the end user has a method to take decisions for the maintenance actions.

This research aims to present a model to identify the component that has failed before its MTTF and, depending on whether or not the cause of the failure is known and the time to restart the normal functioning of the machine, propose the use of a predictive maintenance strategy and further decision-making to ensure the highest possible value from the machine's operating time. For this research, a real case study has been characterized component-by-component, studying the individual maintenance times to obtain the time lost production (TLP) for each component. Figure 1 shows the features of a multistage machine and the conditions on which the proposed maintenance strategies are based.

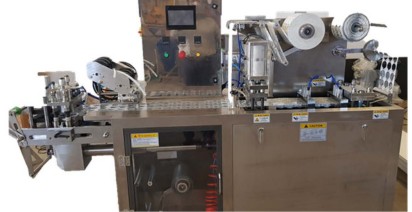

**Figure 1.** Features of a MSTM and main conditions of maintenance policy.

This approach determines the focus of the maintenance strategies, which are always aimed at rapid response, and calculated to avoid unexpected failures, and minimize TLP.

## 2. Materials and Methods

The machine worked for a year with a preventive maintenance system based on the previously characterized components. An algorithm for predictive maintenance was

adopted in the beginning, but only to advise if a component had failed before its MTTF. The authors used a digital behavior twin algorithm [17] for predictive maintenance in this case. A comparison of the components that presented failures before their MTTF is given below. The results allow future users to add predictive maintenance for the components needing supervision to avoid unexpected failures and probable industrial costs for lost production time and quality production.

Below is the methodology used in this research, ordered by steps:

- Step One: The multistage thermoforming machine was selected as the case study. This machine was characterized, and all the components were identified and classified by type. See Section 2.1.
- Step Two: Reliable maintenance times were defined for each component. Importantly, an adequate MTTF was established for each component. See Section 2.2.
- Step Three: Possible preventive maintenance strategies were defined, and predictive maintenance strategies adopted. See Section 2.3.
- Step Four: The components that presented a failure before their MTTF after a year of working were studied. See Section 2.4.
- Step Five: For all of the components, the advice shown by the DBT predictive algorithm was presented to ascertain which failures could be identified before occurring unexpectedly. The advice does not entail a change of maintenance strategy. The only purpose of these dates was for use in data logging. See Section 2.5.
- Step Six: The authors proposed a procedure to make decisions for possible maintenance strategy changes in the components studied by looking for the cause of the failure and then by evaluating two key performance indicators (KPIs). See Section 2.6.

The results, discussion, conclusions, and future research are shown in Sections 3–5.

### 2.1. The Case Study: A Multistage Thermoforming Machine

Thermoforming and tub filling machines are one of many cases, and this study covers this type of machine. Figure 2 shows the MSTM and the placement of the components. The seven steps are identified, together with the main operation in each of them.

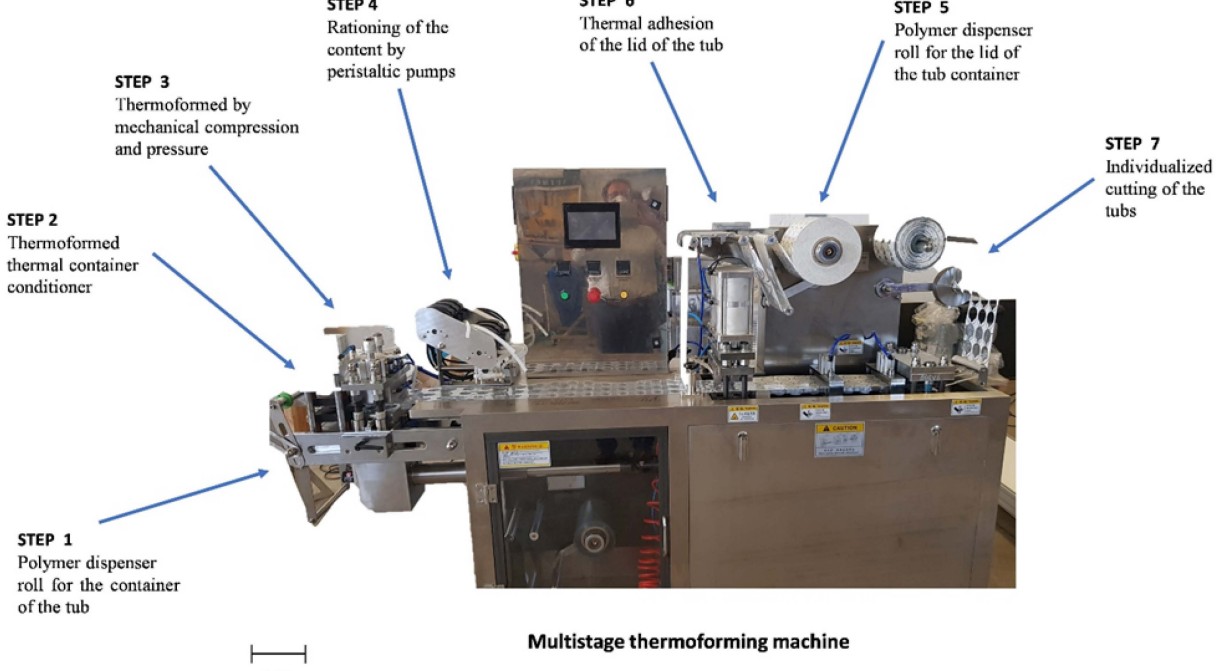

**Figure 2.** A multistage thermoforming machine of six terrines per cycle and its type of components.

This machine has a cycle time of 4 s, and the thermoforming mold allows the manufacture of 6 terrines for each cycle time. So, for each cycle time, seven steps constantly work in coordination.

The proper sequence of steps depends on the programmable logic controller (PLC) inside the electrical panel. The PLC receives all the information provided by sensors and takes decisions for all the actuators at the correct moment.

All the steps may have electrical, electronic, mechanical, and pneumatic components distributed for the whole industrial multistage machine. The adequate state of all the components allows for the correct functioning of the machine and avoids unexpected failures. It is easy to understand that the accumulated work time may affect the state of the components. Due to this and other considerations such as ambient conditions, power supplier events, normal degradation of mechanical components, compressed air system failure or jams in the peristaltic pumping system (step 4), unwanted mechanical shocks can be the origin of unexpected failures in the components and consequently in the industrial multistage machine.

The components of this machine and their type can be seen in Table 1. Many components may have a number greater than one. Also, the figure indicates the possible failure source and the consequences of the failure event.

**Table 1.** List of components in the industrial multistage machine.

| Type of Component | Component | Cause of Failure | Failure Event |
|---|---|---|---|
| Electrical | Master power switch | Ambient condition, Power supplier event | Stop |
| | Plug-in relay | Ambient condition, Power supplier event, Unexpected hit | Malfunction Stop |
| | Command and signalling | Ambient condition, Power supplier event | Stop |
| | Safety limit switch | Ambient condition, Power supplier event, Unexpected hit | Stop |
| Electronic | PLC | Ambient condition, Power supplier event | Stop |
| | HMI | Ambient condition, Power supplier event | Stop |
| | Chromatic sensor | Ambient condition, Power supplier event | Stop |
| | Safety relay | Ambient condition, Power supplier event | Stop |
| | Temperature controller | Ambient condition, Power supplier event, Unexpected hit | Stop |
| | Solid state relay | Ambient condition, Power supplier event | Stop |
| | Belt drive | Ambient condition, Power supplier event | Malfunction |
| | Pressure sensor | Pressure failure, Global fatigue | Malfunction |
| | Servo drive peristaltic pump | Ambient condition, Power supplier event | Stop |
| | Absolute encoder | Global fatigue, Mechanical hit | Malfunction |
| Mechanical | Safety button | Ambient condition, Power supplier event | Stop |
| | Thermal resistance | Ambient condition, Power supplier event | Malfunction |
| | Thermocouple sensor | Global fatigue | Malfunction |
| | Belt motor | Global fatigue | Stop |
| | Bronze cap | Global fatigue | Malfunction |
| | Linear axis | Global fatigue | Malfunction |
| | Linear bearing | Global fatigue | Malfunction |
| | Peristaltic pump | Ambient condition, Power supplier event, compressed air system failure | Stop |
| | Terrine cutter | Global fatigue | Malfunction |
| Pneumatic | Pneumatic valve | Global fatigue | Malfunction |
| | Pneumatic cylinder | Pressure failure, Failure valve | Malfunction |

### 2.2. Maintenance Times for Each Component

Once the industrial thermoforming machine has been characterized, the individual maintenance times required for each component must be studied to adopt the most appropriate preventive maintenance strategy policy accordingly. For this purpose, many individual times and equations are used and presented in this study, which has been provided by J. Jiři [27] and G. Liberopoulos [28].

- TTRP    Time to replace a component
- TTC    Time to configure
- TTMA    Time to mechanical adjustment
- TTPR    Time to provisioning
- MTTR    Mean time to repair
- MTTF    Mean time to failure

- MTBF    Mean time between failure
- TTLR    Line restart time, defined by expert knowledge
- TLP     Time lost production

MTTR (1), TLP (2), and MTBF (3) can be calculated with these equations. Efficiency and availability are used as indicators of success in preventive maintenance.

$$MTTR = TTRP + TTC + TTMA + TTPR \tag{1}$$

$$TLP = MTTR + TTLR \tag{2}$$

$$MTBF = MTTR + MTTF \tag{3}$$

After defining the times and expressions, Table 2 presents the individual maintenance times in seconds for each component in this research. For this machine, the end users and original equipment manufacturer (OEM) have suggested, with their knowledge based on the experience of use, manufacture and maintenance, the fixing of individual maintenance as its shown in Table 2 and global TTLR at 14,400 s.

**Table 2.** Individual maintenance times in s for all the components in the industrial multistage machine.

| Component | MTTR | TTPR | MTTF | TLP |
|---|---|---|---|---|
| Master power switch | 14,400 | 10,800 | 9,999,999 | 28,800 |
| PLC | 435,600 | 345,600 | 9,999,999 | 450,000 |
| HMI | 435,600 | 345,600 | 9,999,999 | 450,000 |
| Chromatic sensor | 176,520 | 172,800 | 5,000,000 | 190,920 |
| Plug-in relay | 14,400 | 10,800 | 5,000,000 | 28,800 |
| Command and signalling | 14,400 | 10,800 | 5,000,000 | 28,800 |
| Safety limit switch | 14,400 | 10,800 | 9,999,999 | 28,800 |
| Safety relay | 14,400 | 10,800 | 9,999,999 | 28,800 |
| Safety button | 14,400 | 10,800 | 9,999,999 | 28,800 |
| Temperature controller | 435,600 | 345,600 | 9,999,999 | 450,000 |
| Solid state relay | 176,400 | 172,800 | 5,000,000 | 190,800 |
| Thermal resistance | 25,500 | 10,800 | 3,700,800 | 39,900 |
| Thermocouple sensor | 14,700 | 10,800 | 3,700,800 | 29,100 |
| Belt drive | 435,600 | 345,600 | 9,999,999 | 450,000 |
| Belt motor | 187,200 | 172,800 | 5,000,000 | 201,600 |
| Bronze cap | 288,000 | 172,800 | 7,750,000 | 302,400 |
| Linear axis | 288,000 | 172,800 | 7,625,000 | 302,400 |
| Linear bearing | 288,000 | 172,800 | 7,500,000 | 302,400 |
| Pneumatic valve | 176,400 | 172,800 | 9,999,999 | 190,800 |
| Pneumatic cylinder | 176,400 | 172,800 | 9,999,999 | 190,800 |
| Pressure sensor | 176,700 | 172,800 | 5,000,000 | 191,100 |
| Servo drive peristaltic pump | 435,600 | 345,600 | 9,999,999 | 450,000 |
| Peristaltic pump | 547,200 | 518,400 | 5,000,000 | 561,600 |
| Terrine cutter | 288,000 | 172,800 | 9,999,999 | 302,400 |
| Absolute encoder | 360,000 | 172,800 | 5,000,000 | 374,400 |

For this type of machine, both components used at the beginning, as well as those that have presented failures, are completely new units, not ones restored by the technical service of each component manufacturer. For necessary components replacements, only in the case of the pneumatic cylinder is it possible to repair the unit by substituting internal components for new components. All other components are replaced by new units.

### 2.3. Maintenance Strategies

In this section, two preventive maintenance strategies are presented, and one predictive maintenance strategy is used:

- Preventive maintenance, based on the MTTF of each component, to avoid unexpected failures during the work process.

- Improved preventive maintenance, based on the above but minimizing the TTPR of each component.
- Digital behavior twin (DBT) for predictive maintenance.

2.3.1. Preventive Programming Maintenance (PPM)

This strategy is based on the MTTF of each component and proposes inspecting and replacing the component once the worked time reaches the MTTF. This is the maintenance strategy adopted for all the components at the beginning of this study.

Once the decision to replace a component is taken, a decision based on its MTTF, lost production time is necessary for the corresponding maintenance operation. As shown by Equation (1), if the MTTR is higher than the value of TTPR, a new maintenance policy can be used to minimize the MTTR. This policy entails an increase in security stocks. Figure 3 shows the ratio TTPR/MTTR in this machine. The values are provided by the machine manufacturers and shown in [17].

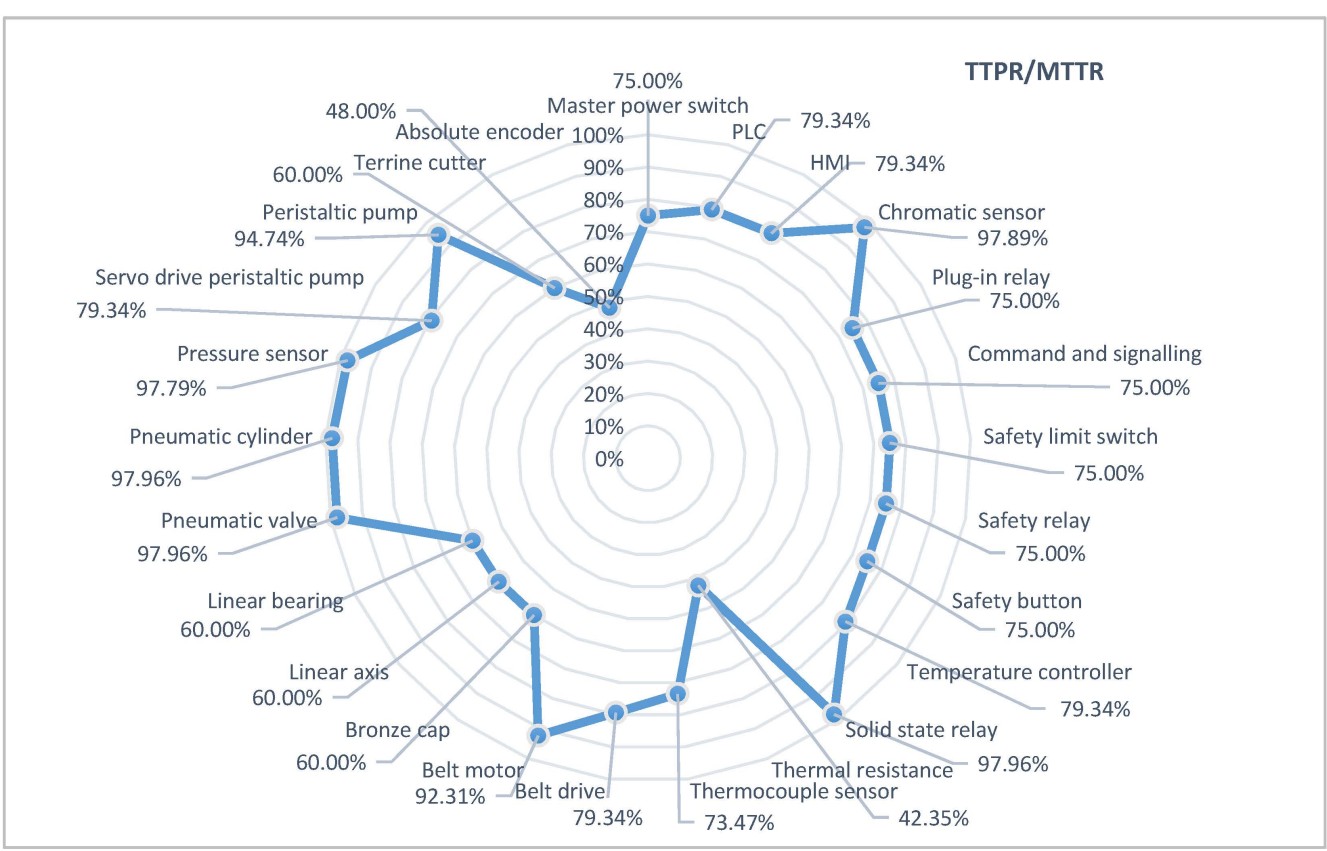

**Figure 3.** Comparison between TTPR and MTTR for the components of the case study.

The significant influence of TTPR value in MTTR is notable. The authors consider this ratio interesting. In Section 2.6, $KPI_1$ and $KPI_2$ will be defined by using TTPR value to propose a change in preventive maintenance strategy.

2.3.2. Improved Preventive Programming Maintenance (IPPM)

This strategy is based on the PPM strategy. When a component has a higher value of TTPR, this strategy can be used to minimize the TLP of the industrial multistage machine. In this case, the TTPR is replaced by a residual time fixed in 300s, which is the time it takes the end user of the machine to collect it from its replacement stock. Garcia, F.J.Á. and Salgado, D.R. [11] proposed a matrix to decide on the most appropriate preventive maintenance strategy but not on the component that needs a predictive maintenance strategy.

### 2.3.3. Digital Behaviour Twin (DBT)

This strategy is based on the sensors placed on the machine. These sensors give their values to a PLC, and the PLC uses an algorithm that triggers maintenance recommendations to avoid unexpected failures. A human interface machine (HMI) is used to show these recommendations.

This algorithm uses the signals received from the sensors and refers them to the position of a central axis by means of an absolute encoder. Since the normal operating condition is known, the algorithm detects normal operation without the need for learning, provides warnings of possible faults, and can provide the number of work cycles performed without faults.

Figure 4 shows the conceptualization of this strategy in the cited industrial multistage machine.

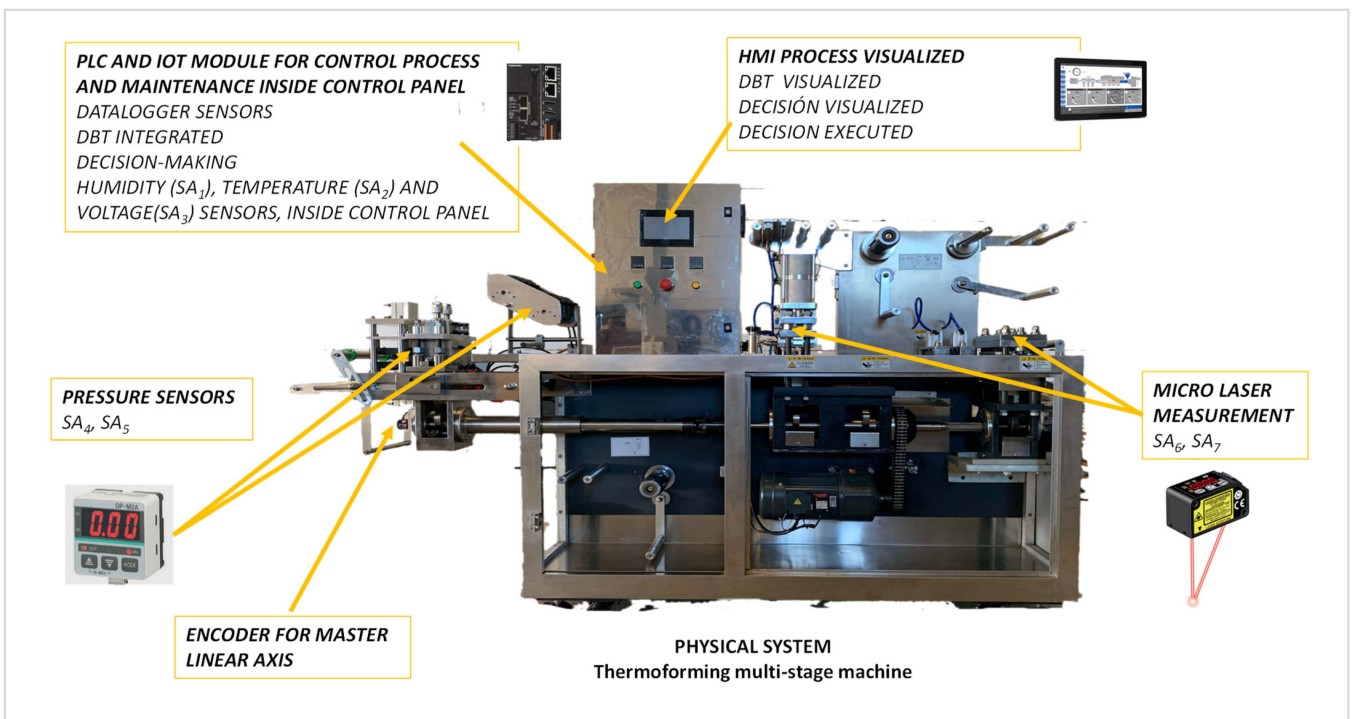

**Figure 4.** Conceptualization of the DBT predictive maintenance strategy.

Garcia, F.J.Á. and Salgado, D.R. [17] described this predictive maintenance strategy in detail. The objective of their research was not to define the predictive algorithm but to use it to propose a method to decide on a change of maintenance strategy from preventive to predictive.

This strategy has already been tested in this industrial multistage machine and allows the detection of potential failures within each cycle of operation of the whole machine.

### 2.4. Recovered Data after a Year of the Machine Working

The industrial multistage machine worked without stopping, 8 h per day, Monday to Friday, for one year, with the PPM in place and the DBT functioning only for data logging advice. Table 3 shows the list of components with the corrected MTTF if the component had failed before its MTTF and whether the cause of the failure was known or unknown.

**Table 3.** Individual component failures occurring within one year of the machine's working. Times in s.

| Component | Fails before MTTF | Cause of Failure | Corrected MTTF |
|---|---|---|---|
| Chromatic sensor | 1 | Known | 3,998,750 |
| Plug-in relay | 1 | Known | 4,056,010 |
| Temperature controller | 1 | Known | 7,934,710 |
| Solid state relay | 1 | Known | 4,678,034 |
| Thermal resistance | 1 | Unknown | 3,067,090 |
| Thermocouple sensor | 1 | Unknown | 2,890,760 |
| Bronze cap | 1 | Unknown | 6,500,453 |
| Linear bearing | 1 | Unknown | 6,375,010 |
| Pressure sensor | 1 | Unknown | 4,575,102 |
| Peristaltic pump | 1 | Known | 4,434,090 |
| Terrine cutter | 1 | Unknown | 8,750,778 |
| Absolute encoder | 1 | Known | 4,756,002 |

Table 3 shows that many components have presented failures before their MTTF. Figure 5 shows the results by type of component. The pneumatic components have not presented failures before their MTTF, unlike the rest of the components.

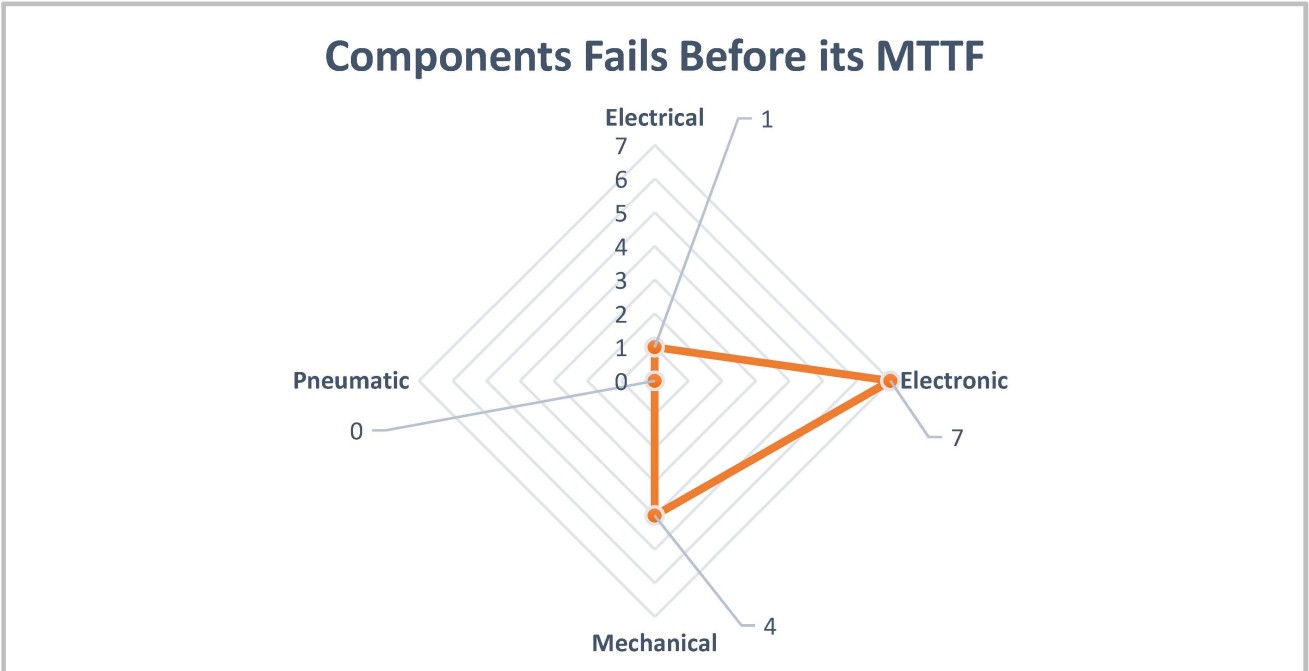

**Figure 5.** Component failures before their MTTF.

Table 4 shows the description of the cause of the failure of the components that presented a failure before their MTTF and also indicates, for these components, whether the cause is due to an occasional (infrequent) situation or a normal situation.

In the case of the plug-in relay, the authors believe that this component was not completely new at the beginning of the experiment. To verify this, a quality test was carried out. The rest of the components shown in Table 4, presented a failure-for-occasional-situation. In Section 2.6 a procedure to avoid the same situation is proposed.

These failures were registered. The DBT algorithm used only for data logging the advice for an unexpected failure will be compared below.

**Table 4.** Registered known causes of failure in components that presented a failure before their MTTF.

| Component | Situation | Description of the Known Cause of Failure |
|---|---|---|
| Chromatic sensor | Occasional situation | The supplier of the film for the terrine lid changed the color without prior notice and made it darker and more reflective. This caused the sensor to stop seeing the mark correctly. |
| Plug-in relay | Normal situation | The number of commutations exceeded the mechanical endurance. |
| Temperature controller | Occasional situation | Mixed events of voltage RMS and high level of humidity. |
| Solid state relay | Occasional situation | The higher level of humidity and air dust caused a short circuit. |
| Peristaltic pump | Occasional situation | A higher density of fluid dosed in the terrine caused a jam. |
| Absolute encoder | Occasional situation | Accidental mechanical shock. |

*2.5. DBT Predictive Algorithm Warnings of Failure Recovered*

The DBT algorithm matched the real failures that occurred when the machine was working. Table 5 shows the warning of failures obtained for the DBT predictive maintenance strategy. This table only shows components that presented failures.

**Table 5.** DBT warnings of failures by components within the operation time studied.

| Component | DBT Warning of Failures |
|---|---|
| Chromatic sensor | 1 |
| Plug-in relay | 1 |
| Temperature controller | 1 |
| Solid state relay | 1 |
| Thermal resistance | 1 |
| Thermocouple sensor | 1 |
| Bronze cap | 1 |
| Linear bearing | 1 |
| Pressure sensor | 1 |
| Peristaltic pump | 1 |
| Terrine cutter | 1 |
| Absolute encoder | 1 |

The coincidence of warning of failures provided by DBT and shown in Table 5 and components that failed before their MTTF, as shown in Table 3, suggests using the DBT algorithm if a component requires predictive maintenance. Due to this coincidence, a scorecard must be designed to make decisions about changes in the component maintenance strategy.

*2.6. Method Proposal to Take Decisions for Maintenance Strategy Decisions*

As Section 2.3.1 set forth, the PPM strategy had been adopted for all the components at the beginning of the study. With all the accumulated dates compared to the actual event, and the warning advice given by the DBT algorithm, this section explains how to make decisions for a possible change of maintenance strategy.

The objective is to identify the components that need predictive maintenance. For this purpose, there are two key questions:

- Has the component failed before its MTTF?
- Do we know why it failed?

Two key performance indicators are studied to ascertain whether the reason is known. The expressions for $KPI_1$ (4) and $KPI_2$ (5) are the following:

$$KPI_1 = (MTTR - TTPR)/MTTR \tag{4}$$

$$KPI_2 = TTPR /TLP \tag{5}$$

$KPI_1$ is used to ascertain the influence of TTPR in the MTTR for each component. If this ratio presents a small value, the TTRP will be higher, which is considered an important piece of information with reference to changing the maintenance strategy.

$KPI_2$ is used to assess the influence of TTPR in Time TLP because this ratio shows the availability and efficiency decrease for a higher value of TTPR.

### 2.6.1. Procedure to Set KPI$_1$ and KPI$_2$ Values

The procedure to set initial values of KPI1 and KPI2 is the following:

- Calculate KPI1 interval between PPM and IPPM strategies.
- Calculate KPI2 interval between PPM and IPPM strategies.
- Calculate average value of KPI1 and KPI2, assuming PPM strategy.
- Calculate average value of KPI1 and KPI2, assuming IPPM strategy.
- Calculate average value of TTPR/MTTR ratio assuming PPM strategy.

Individual times for PPM strategy are shown in Table 2. In the case of IPPM strategy, only TTPR is modified for a constant value fixed in 300s (see Section 2.3.2).

Table 6 shows all of the calculated values.

**Table 6.** Calculated ratios to define fixed values of KPI$_1$ and KPI$_2$.

| Strategy | Ratio | Average KPI$_1$ | Average KPI$_2$ | Average TTPR/MTTR |
|---|---|---|---|---|
| PPM | Value | 22.92% | 63.14% | 77.08% |
| | Interval | [2.04–57.65%] | [27.07–92.31%] | |
| IPPM | Value | 96.04% | 0.99% | 3.96% |
| | Interval | [92.31–99.84%] | [0.15–1.64%] | |

As can be observed, the intervals for KPI$_1$ and KPI$_2$ values in PPM and IPPM strategies have no common points. For initial, fixed KPI's points, we must be within the intervals provided by PPM strategy. Whether KPI$_2$ is considered the average ratio between TTRP and MTTR due to the TLP depends on a constant value (TTLR equal to 14,400 s) and the MTTR value (see Equation (2))

Figures 6 and 7 show the fixed KPI$_1$ and KPI$_2$ values for decision-making. A large dispersion of KPI$_1$ and KPI$_2$ values is observed in the case study. The correct functioning of the fixed values is evaluated by the minimization of TLP and stock cost in case of adopting IPPM strategy. The final value fixed in the case of KPI$_1$ is 25% and in the case of KPI$_2$ is 70% (value obtained by comparing average KPI2 with average TTPR/MTTR).

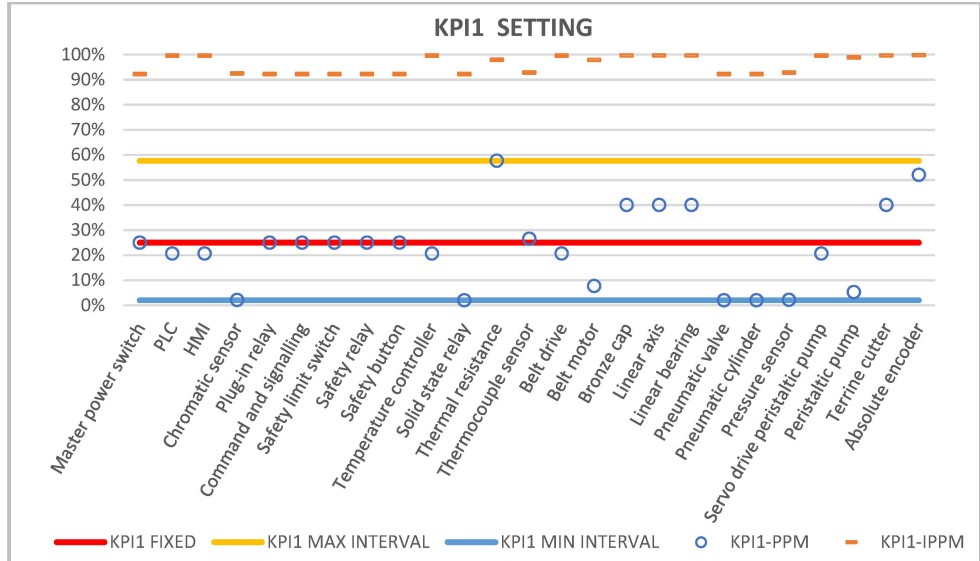

**Figure 6.** Comparison of KPI$_1$ values for each component in PPM and IPPM strategies, KPI$_1$ interval in PPM strategy and fixed value 25% of KPI$_1$.

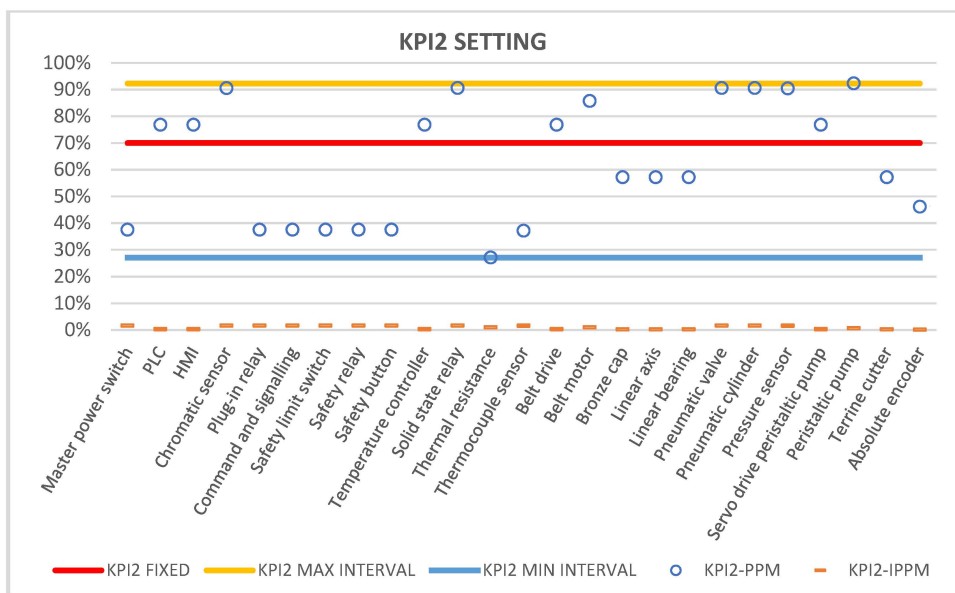

**Figure 7.** Comparison of KPI$_2$ values for each component in PPM and IPPM strategies, KPI$_2$ interval in PPM strategy and fixed value 70% of KPI$_1$.

With the fixed values, decision-making to change maintenance strategy can be adopted. So:

- If KPI$_1$ < 25% and KPI$_2$ > 70%, the improved preventive maintenance strategy can be proposed, with a previous GOC evaluating the component;
- If KPI$_1$ > 25% and KPI$_2$ < 70%, a preventive maintenance strategy change is unnecessary.

Of course, if a component presents a failure before its MTTF and the cause of the failure is unknown, and the value of KPI$_1$ < 25% and KPI$_2$ > 70%, several changes must be made in the maintenance strategy.

### 2.6.2. Proposed Method for Maintenance Strategy Adoption

Figure 8 shows the method proposed by a PPM strategy adopted for all the components of the industrial multistage machine initially and the possible decisions to be taken depending on the knowledge of the fault and the KPI values.

The proposed method can be used in this machine when assessing whether to change the maintenance strategy if a component fails before its MTTF. However, feedback is useful to control the real evolution of the machine in every possible way. The authors consider that this feedback is only useful if the cause of the failure is known.

With the proposed method, all the components start operating with a PPM strategy, and if any fail before their MTTF, a change to IPPM or IPPM with DBT may be appropriate.

As mentioned in Section 2.5, if it is necessary to use a predictive maintenance strategy, DBT can be used, due to the good results offered with the advice shown in Table 5. In this way if a component fails before its MTTF, and the cause is unknown, IPPM will be adopted regardless of the value of KPIs. Later, an inspection of the location and other factors to find the cause of the failure with DBT monitoring enables a new value of MTTF to be set. If the cause of the failure is found, the method returns to starting point. If the component does not fail before its new MTTF, the maintenance strategy will be PPM. Otherwise, the way depends on the knowledge of the second failure.

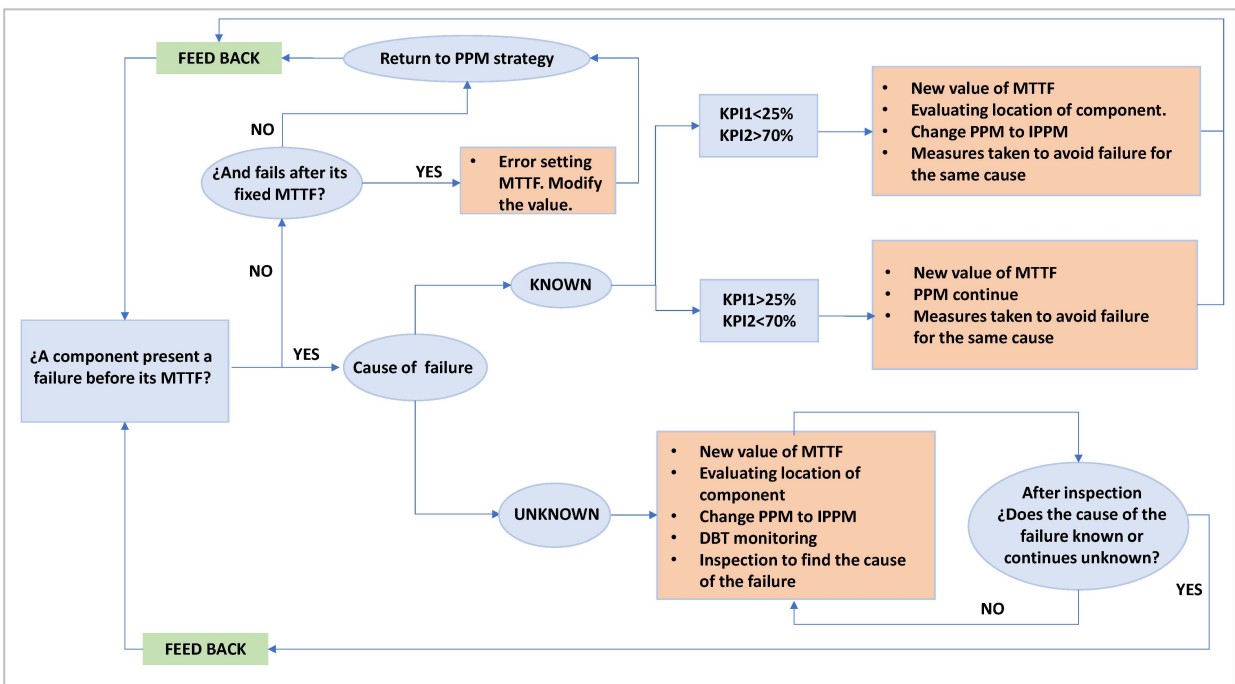

**Figure 8.** Proposed method to switch from a preventive maintenance strategy to a predictive strategy with a DBT algorithm.

According to the proposed method it i considered that the principal object and benefit of adopting a DBT predictive maintenance strategy is to determine the reason of a component failure before its MTTF by using distributed sensors that are part of the DBT predictive maintenance strategy. Also, when DBT monitoring is used, the behavior of the whole machine is monitored to guarantee the correct functioning of the MSTM. So, as Figure 5 shows, if the cause of the failure is found, the replaced component returns to the starting point of the proposed method and the DBT monitoring continues until the next failure of the same component. Otherwise, the method does not continue towards the starting point, and the model waits instead for a better result of the main actions proposed to find the reason of the failure.

The final object of this dynamic method is the default selection of PPM strategy for all components, and only IPPM if KPI1 < 25% and KPI2 > 70% at the same time if these components fail before its MTTF. Nevertheless, the real situations involving these types of machines, located in important factories and parts of a production process, indicates to us the need for a predictive maintenance strategy if, due to an occasional (infrequent) situation, a component fails before its initial MTTF.

### 3. Results

The application of the proposed method for possible maintenance strategy changes is shown in Table 7. The initial maintenance strategy was PPM for all the components. The next column shows the maintenance strategy to be adopted if the component fails before its MTTF.

The results show different changes in maintenance strategies. The PPM strategy of all the components that failed before their MTTF due to unknown causes was changed to IPPM with DBT monitoring (pressure sensor, thermal resistance, thermocouple sensor, bronze cap, linear bearing, and terrine cutter). The maintenance strategy of two components with known causes of failure remained unchanged (plug-in relay and absolute encoder), and the maintenance strategy of four components (chromatic sensor, temperature controller, solid state relay, and peristaltic pump) was changed to IPPM.

Of course, the initial PPM strategy of components that did not fail remained the same.

**Table 7.** Maintenance strategies for all components after a year in operation using the proposed method.

| Component | $KPI_1$ | $KPI_2$ | Maintenance Strategy after a Year of Work |
|---|---|---|---|
| Master power switch | 0.25 | 0.38 | PPM |
| Plug-in relay | 0.25 | 0.38 | PPM |
| Command and signalling | 0.25 | 0.38 | PPM |
| Safety limit switch | 0.25 | 0.38 | PPM |
| PLC | 0.21 | 0.77 | PPM |
| HMI | 0.21 | 0.77 | PPM |
| Chromatic sensor | 0.02 | 0.91 | IPPM |
| Safety relay | 0.25 | 0.38 | PPM |
| Temperature controller | 0.21 | 0.77 | IPPM |
| Solid state relay | 0.02 | 0.91 | IPPM |
| Belt drive | 0.21 | 0.77 | PPM |
| Pressure sensor | 0.02 | 0.90 | IPPM + DBT monitoring |
| Servo drive peristaltic pump | 0.21 | 0.77 | PPM |
| Absolute encoder | 0.52 | 0.46 | PPM |
| Safety button | 0.25 | 0.38 | PPM |
| Thermal resistance | 0.58 | 0.27 | IPPM + DBT monitoring |
| Thermocouple sensor | 0.27 | 0.37 | IPPM + DBT monitoring |
| Belt motor | 0.08 | 0.86 | PPM |
| Bronze cap | 0.40 | 0.57 | IPPM + DBT monitoring |
| Linear axis | 0.40 | 0.57 | PPM |
| Linear bearing | 0.40 | 0.57 | IPPM + DBT monitoring |
| Peristaltic pump | 0.05 | 0.92 | IPPM |
| Terrine cutter | 0.40 | 0.57 | IPPM + DBT monitoring |
| Pneumatic valve | 0.02 | 0.91 | PPM |
| Pneumatic cylinder | 0.91 | 0.91 | PPM |

## 4. Discussion

The proposed method for changing the maintenance strategy for all the components that failed does not provide a static decision criterion. For example, the same component can fail first due to an unknown cause, and again a second time due to a known cause. The method allows taking different decisions according to whether or not the cause of the failure is known.

The authors consider knowing the cause of the failure critical. An industrial multistage machine must not operate with unknown failures. Also, once the cause is known, the manufacturer must take action to avoid an unexpected failure due to the same cause. If these actions are correct and there is feedback, the industrial multistage machine can restart operating with adequate functionality guarantees.

The values of $KPI_1$ and $KPI_2$ are used to assess whether a change of preventive maintenance strategy is required. As mentioned in Section 2.6, the extreme values of both are fixed to show whether the preventive maintenance strategy should be changed from PPM to IPPM. However, if the value of time to provisioning (TTPR) of a component goes up or down, the value of its $KPI_1$ and $KPI_2$ will also change. In this scenario, if a failure occurs in this component, the method will use another way to make decisions, in a further evaluation.

A continuous application of this method for the same industrial multistage machine will allow greater failure control and higher levels of operation time without failures.

If a component supplier is changed for market reasons, the proposed method must be reassessed, and the KPIs and MTTR must be recalculated. Also, the value of MTTF must be changed accordingly before the machine resumes its operation.

The authors consider the following assessment critical, ascertaining the trust level in the component manufacturer by evaluating the ratio between the corrected and initial MTTF of all the components that failed before their initial MTTF. Table 8 shows this ratio:

**Table 8.** Trust levels in component manufacturers by comparing the initial and corrected MTTF in components that failed before their initial MTTF.

| Type of Component | Component | Trust Level |
|---|---|---|
| Electronic | Chromatic sensor | 79.98% |
| Electrical | Plug-in relay | 81.12% |
| Electronic | Temperature controller | 79.35% |
| Electronic | Solid state relay | 93.56% |
| Electronic | Thermal resistance | 82.88% |
| Electronic | Thermocouple sensor | 78.11% |
| Mechanical | Bronze cap | 83.88% |
| Mechanical | Linear bearing | 85.00% |
| Electronic | Pressure sensor | 91.50% |
| Mechanical | Peristaltic pump | 88.68% |
| Mechanical | Terrine cutter | 87.51% |
| Electronic | Absolute encoder | 95.12% |

This assessment, therefore, allows for a long operating time with an adequate selection of component manufacturers. Figure 9 shows the average trust level (ATL) by component type. The authors consider that the compared values must be similar. This indicates that the machine maintenance team is adequate for the whole machine. Obviously, the optimal value of this ATL is 100%. As the pneumatic components did not fail before their MTTF, they have not been included in Figure 9.

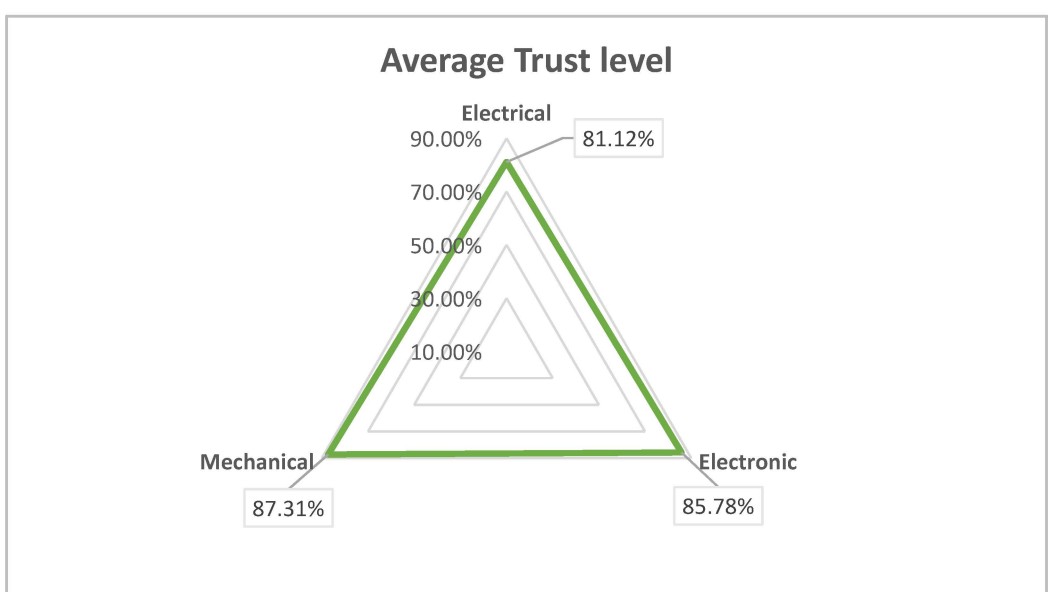

**Figure 9.** Average Trust level of components that failed before their initial MTTF.

The authors consider that one way to further this research would be if an ATL were to be fixed for all the components for possible decisions to change component manufacturers. Also, a new result would be obtained if the cost of the component were to be used in this proposed future research.

## 5. Conclusions

The proposed method for possible maintenance strategy changes for components in the same industrial multistage machine provides ways to change the maintenance strategy for PPM to IPPM or IPPM with DBT monitoring. The authors consider that this method will be useful for other industrial multistage machines.

The predictive maintenance strategy is used for constant component monitoring if an unexpected failure has occurred, so if the cause is known and the measures for avoiding a

new failure for the same cause are taken, the component will probably fail at its new MTTF and will then go back to having preventive maintenance like PPM or IPPM.

If a component presents many consecutive failures before its initial MTTF and the actions proposed by the method are taken, then the ATL of the component manufacturer should be revisited to decide on whether to change the manufacturer with an objectively higher quality in this component. In this case, this could be another way to start a failure mode and effect analysis (FMEA) to redesign the function, location, and work of this stage of the machine.

All the components can be included in the study of ATL by component type. In this case, the average value will be higher than that shown in Figure 9. The authors only included the components that failed to avoid wrong results in the machine maintenance team's evaluation.

As the trust level, or the ATL, depends on the ratio between initial MTTF and real MTTF of the component or type of components, these values do not improve with a maintenance strategy change; they only improve to 1 or 100% if the actions necessary to take for avoid occasional (infrequent) situations that end with an unexpected failure work correctly. In the case of trust level or ATL improvement up to 100% the authors suggest an incorrect initial MTTF fixed at starting point of the machine's use. The proposed method suggests this way (see Figure 8)

Industrial multistage machines need a long working time without unexpected failures, so a global method for taking the appropriate decisions for maintenance strategies is needed, and adequate changes must be made to avoid such unexpected failures. The proposed method allows reaching this objective. Nevertheless, some comments for its application in the context of other multistage machines must be related:

- The case study is a multistage thermoforming machine. This machine has an absolute encoder. Its position is constantly sent to the PLC for synchronization and management of all the coordinated steps in the correct order. This encoder allows the use of a digital behavior twin algorithm for predictive maintenance strategy. Not all of the multistage machines have an encoder for this special function, so the normal behavior of the machine must be referred to with a more precise physical analogue.
- Due to the fact that the cycle time is only 4 s, the algorithm for predictive maintenance must be speedy and certain. Other machines with longer cycle times could use predictive maintenance based on the time;
- As Figure 1 indicates, the preventive maintenance strategy depends upon the individual maintenance times. It would be interesting to evaluate the sensibility of the method for an incipient change of TTPR in some components due to global market conditions.

The main contributions highlighted in this article are:

- Providing a method for deciding when to use predictive maintenance strategy and when to stop it in different components of a MSTM.
- Providing a dynamic global method to establish the maintenance strategy of any component of an MSTM.
- Providing a confidence level of a component or type of components in an MSTM that indicates whether the MTTF of said component operating in said machine is reliable.
- Because of the above, obtaining information on the reliability of the components of a MSTM to avoid unexpected failures during its operating time.

Table 9 shows the results of the comparison between the introduction citations and the proposed method. Due to the singularity of this type of multistage machine, the cited references are not alternative methods that can be used to provide other maintenance strategies for the same machine in the same working conditions, with the same components and the same evaluation time (1 year). Due to this, the comparison offered in the following table focuses on the most significant aspects found in each citation that are related to the methodology developed in this study. This comparison is, therefore, in qualitative terms, and not able to offer numerical comparisons. The first column indicates the item or relevant

aspect to be compared. The second column indicates the highlighted references compared. The third column is a qualitative comparison between the cited item (column 1) in the reference (column 2) and the method proposed in this article.

**Table 9.** Qualitative comments highlighted between the proposed method and the state of art. (*) Improve options.

| Item | References | Qualitative Comments after Comparison |
|---|---|---|
| Minimizing security stocks | [1,2] | Correct selection of fixed KPIs allows the optimization of stock and provides the adequate preventive maintenance policy |
| Stops to settings, removal actions. Imperfect maintenance | [3,4] | Settings only at the start time of the machine functioning by the temperature controller, thermal resistance, and thermocouple sensor. The maintenance actions must perform the machine functioning. The system can evaluate if the actions in each component or each type of component are imperfect by trust level or ATL. |
| Mathematical model for Preventive maintenance | [5–8] | Complex, very theoretical and many variables to manage. Simple, sensitive to variations of individual maintenance times. |
| MTTF reliable Reliability and law degradation | [9–11] | Initial MTTF fixed for all components; reliability functions not used. Possibility to change MTTF value if real MTTF lower or upper than initial MTTF fixed. |
| Product design and operation conditions | [12] | If a component exhibits repeated failures, an immediate FMEA analysis procedure is initiated to find design errors or component selection errors. |
| Mathematical model for Predictive maintenance | [13–18] | Uses PLC with embedded DBT algorithm. No need training and learning time. Quick response Very useful for a machine with fast cycle time. |
| Location components | [19] | (*) Possible improvement. Can be evaluated for this application |
| Mixed cost and technical analysis | [20,21] | (*) Possible improvement. Also is cited in future research. Coincidence in the use of FMEAS analysis |
| Digital Twin | [22,23] | The behavior of the machine always is the same and does not need a real digital twin since the characterization is special for each MSTM and operation conditions are always are the same. Coincidence in the event failure advises, no training and utilization of FMEAS analysis. |
| Augmented Reality and Computer Vision | [24,25] | (*) Possible improvement. Not used. ATL is used for evaluating the maintenance operator actions required for maintenance policy. But it is used after a maintenance action. |
| Preventive actions in flexible windows time. Predictive maintenance always running Method for decision-making | [26] | (*) Possible improvement to use flexible windows time for preventive maintenance actions. Predictive maintenance only works if a component fails before its MTTF, and the cause of the failure is unknow. Coincidence in the contribution of a method for decision-making |
| Individual preventive maintenance Times | [27,28] | Used in the article and performed by developing KPIS for preventive maintenance decisions |

The method proposed is appropriate for the MSTM but can improve with respect to some items.

Future research:

- Study the influence of a fixed ATL and cost assessment for possible component manufacturer changes;

- Utilization of DBT monitoring for combined supervision in parallel of the same machine system to use Predictive Maintenance and use the advice for one machine to start DBT monitoring in other machines of the system working in the same operating conditions;
- Global cost analysis of the components, DBT monitoring system, and their influence on possible maintenance strategies for all the components in an industrial multistage machine;
- Mixed method for maintenance strategies using technical parameters and cost terms.

**Author Contributions:** Conceptualization, F.J.Á.G. and D.R.S.; methodology, F.J.Á.G.; software, F.J.Á.G.; validation, D.R.S. and F.J.Á.G.; formal analysis, F.J.Á.G.; investigation, F.J.Á.G. and D.R.S.; resources, F.J.Á.G.; data curation, F.J.Á.G.; writing—original draft preparation, D.R.S. and F.J.Á.G.; writing—review and editing, F.J.Á.G. and D.R.S.; visualization, F.J.Á.G.; supervision, D.R.S.; project administration, D.R.S.; funding acquisition, D.R.S. and F.J.Á.G. All authors have read and agreed to the published version of the manuscript.

**Funding:** This study has been carried out through the Research Project GR-21098 linked to the VI Regional Research and Innovation Plan of the Regional Government of Extremadura.

**Data Availability Statement:** Not applicable.

**Acknowledgments:** The authors wish to thank the European Regional Development Fund "Una manera de hacer Europa" for their support of this research. This study has been carried out through the Research Project GR-21098 linked to the VI Regional Research and Innovation Plan of the Regional Government of Extremadura.

**Conflicts of Interest:** The authors declare no conflict of interest.

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
