# Peer review of "An Approach for Predictive Maintenance Decisions for Components of an Industrial Multistage Machine That Fail before Their MTTF: A Case Study"

_systems, doi:10.3390/systems10050175_

Round 1

Reviewer 1 Report

This study developed a model to identify the component that has failed before its mean time to failure (MTTF). The authors propose the use of a predictive maintenance strategy and further decision-making to ensure the highest possible value of the machine’s operating time. In this study, a real case study has been characterized component by component with the individual maintenance times to obtain the time lost production (TLP) for each component. It is a very interesting research. However, some general critical points need to be clarified before it is published.

Main comments:

1.      In Table 2, the maximum MTTF is 9999999. It is about one year if these components work 8 hours per day. How to obtain the MTTF? How to ensure all the same type components have the same MTTF?

2.      In Table 2, the components used for experiments is whole new components or overhaul components? I think it is very different for these two types of components. Actually, in production line, most components are overhauled. How to apply the results on a real production line?

3.      According to Figure 3 and Table 4, these components failure before their MTTF is a normal situation or occasional situation? If the causes of failure in a component are different, are the MTTF of a component the same?

4.      The title of this study is “An approach for predictive maintenance decisions for components ...” From my viewpoint, this study is not for predictive maintenance decisions. It is only for maintenance decisions. The authors used KPI1 and KPI2 to decide whether the maintenance strategy should be changed.

5.      The authors did not describe that any benefit can obtain if the maintenance strategy changed.

6.      The authors did not explain why the authors consider the critical values of 25% for KPI1 and 70% for KPI2. Are these two values suitable for all types of components?

7.      According to Figure 4 and Table 6, it indicates that when Kpi1>70% and Kpi2<25%, the maintenance strategy changes PPM to IPPM. We only find 6 components change PPM to IPPM+DBT monitoring, but their Kpi1 and Kpi2 do not satisfy the criterion, why? For example, the Kpi1=0.02 and Kpi2=0.9 for pressure sensor.

8.      In Table 7, the trust level can be improved or not if we change the maintenance strategy?

Minor comments:

1.      MTTF indicates “Mean Time To Failure”. I found some typo errors in abstract and in the main text. Please check and revise the manuscript carefully.

2.      Line 24, the citation style is not correct to cite a reference in the main text, please check the MDPI Citations Style Guide and revise the main text.

3.      Line 151, it needs to explain DBT when you first time mention the abbreviation.

4.      In Figure 4, there is a encoding error in the description: ??? component present a failure between MTTFi. What is MTTFi?

Author Response

We have just finished the review of our paper (Manuscript ID: systems-1845135) titled “An approach for predictive maintenance decisions for components of an industrial multistage machine that fails before their MTTF. A case study”. We sincerely appreciate the suggestions of the reviewers. We have tried to comply all the comments, which have been used to improve the quality of the original manuscript.

Reviewer 2 Report

Dear authors,

the submitted work tries to capture a really interesting task that can be generalized to several industries by adopting or modifying the corresponding values / attributes that characterize each different domain. However, there are serious flaws in both the scientific part and the structure that you followed.

- You do not even describe the overall problem, but start posing related work from literature

- The main contributions of that work are not explicitly highlighted

- Several lengthy sentences are used, while several syntax and punctuation issues are met across the manuscript. Please be more careful with those issues (e.g., A Gharbi -> A. Gharbi)

- The visualization part is poor, and the information that you need to present through tables include several redundant recordings (e.g., the whole Table 5 and the 4th column of Table 6)

- There is no clear understanding how your method outreaches the rest ones, and especially those that are data-driven.

Author Response

(The authors gave the same response as above.)

Reviewer 3 Report

The article is relevant, very interesting and deals with a current issue.

- The abstract is clear and seems to have all the necessary elements.

- The introduction section gives a good framework to the problem and presents also the literature review with explanation of the main concepts needed.

- Regarding the section 2 - Materials and Methods - some of the content presented there should be moved to the results. This section should be devoted to explaining the methods used exclusively, leaving the results to the next section. In this case, it is suggested that the authors divide this section.

- In the conclusions, the authors should highlight the implications of the study, as well as theoretical and practical contributions

Author Response

We have just finished the review of our paper (Manuscript ID: systems-1845135) titled “An approach for predictive maintenance decisions for components of an industrial multistage machine that fail before their MTTF. A case study”. We sincerely appreciate the suggestions of the reviewers. We have tried to comply all the comments, which have been used to improve the quality of the original manuscript.

Reviewer 4 Report

The authors have presented an approach for predictive maintenance decisions for components of an industrial multistage machine that fail before their MTTF. Its an interesting work. The reviewer has the following comments. 

1) In abstract it is unclear about the actual system to be monitored. 

2) Introduction: Need to be revised completely. There is no connectivity between literature works. It looks like a separate statements. 

3) DBT algorithm need to discussed in detail with mathematical formulations. 

4) Figure 4 can be improved in a better way to understand it by everyone. 

5) list the contributions of the paper with respect to the state of the art. 

6) What are the limitations of the proposed work. 

7) Need more discussion on fault conditions. 

8) Plot the normal and fault condition signals. 

Author Response

(The authors gave the same response as above.)

Round 2

Reviewer 1 Report

Journal: Systems

Article: systems-1845135-peer-review-v2

Title: An approach for predictive maintenance decisions for components of an industrial multistage machine that fail before their MTTF. A case study.

The authors have responded to the review’s comments. But I find some mistakes are not revised even though the authors reply that they have done. In addition, some critical points about preventive maintenance and predictive maintenance need to be clarified before it is accepted.

 Main comments:

1.      Line 245, I think MTTR is always higher than the value of TTPR unless TTRP= TTC=TTMA=0 from eq. (1). Right? The values of TTRP, TTC, and TTMA are always larger than zero, so MTTR must be higher than the value of TTPR.

2.      I am confused by the description in line 345 and Figure 4. In line 345, it indicated that the criterion is “KPI1<25% and KPI2> 70%” and “KPI1>25% and KPI2< 70%”, but the criterion is “KPI1> 70% and KPI2<25%” and “KPI1<70% and KPI2>25%” in Figure 4. Which one is correct?

3.      As mentioned previously, MTTF indicates “Mean Time To Failure”. But I find the authors did not revise some typographical errors in the main text.

4.      As mentioned previously, in Figure 4, there is a encoding error in the description, as shown in the figure above.

5.      The authors still did not explain why these two critical values of 25% for KPI1 and 70% for KPI2. I suggest the authors provide a detailed description of how to decide these two values.

6.      The author described that the definition of Average Trust level (ATL) is equal to the ration of Corrected MTTF and Original MTTF, right? What is the ATL if the component did not fail after a year of the machine working? Will the ATL be larger than 100%? What does it mean when ATL > 100%?

7.      I think the most important issue is to define the decision-making rule for predictive maintenance.

8.      According to Figure 4 and Table 6, it indicates the criterion of the two key performance indicators. Why not adopting predictive strategy with a DBT algorithm for all components? Thus, we can avoid all unexpected failures in production line.

Minor comments:

1.      Line 241, what is “MMTF”?

2.      I suggest that the heading of 2.3.1 and 2.3.2 should be “Preventive programming maintenance. PPM” and “Improved preventive programming maintenance. IPPM” according to reference 11.

3.      Line 305, this paragraph needs grammatical revision.

Author Response

Dear Reviewer.

We have just finished the review of our paper (Manuscript ID: systems-1845135) titled “An approach for predictive maintenance decisions for components of an industrial multistage machine that fails before their MTTF. A case study”. We sincerely appreciate the suggestions of the reviewers. We have tried to comply all the comments, which have been used to improve the quality of the original manuscript.

Reviewer 2 Report

Dear authors,

thank you for your answers. Your applied modifications seem to have improved the total manuscript. However, as i stated before, there are several and important flaws:

- The Section 1 is incompatible with the structure of a scientific journal. You just pose references in separate paragraphs that are never later discussed or posed in contrast to your proposed work. 

- Although you tried to highlight the ambition of that work, I cannot see any comparisons or relation with the existing literature. Can this problem be solved just with some hand-crafted rules? Did you compare them with any data-driven method? All those results seem to characterize an ideal case, while the extraction of results from just one sample is not a strong proof. What did other approaches use into their comparisons?

- Some visualizations and data summarization still face problems.

There are not any strong points for proving your assumptions.

Author Response

(The authors gave the same response as above.)

Reviewer 4 Report

the revised version is satisfactory

Author Response

(The authors gave the same response as above.)

Round 3

Reviewer 1 Report

The authors have provided responses to the review’s comments. I recommend accepting this paper after minor revision.

Minor comments:

1.      This is the third time to mention that MTTF indicates “Mean Time To Failure”. But I find in the Abstract, Keywords and main text, there are some typo errors “Main Time To Failure”. Please check this carefully.

2.      Line 403, it should be “In case ...” not “I case ...”

Author Response

We have just finished the review of our paper (Manuscript ID: systems-1845135) titled “An approach for predictive maintenance decisions for components of an industrial multistage machine that fail before their MTTF. A case study”. We sincerely appreciate the suggestions of the four reviewers. We have tried to comply all the comments, which have been used to improve the quality of the original manuscript.

Reviewer 2 Report

Dear authors,

your work as it concerns the presentation style and the depth of the research has been slightly improved.

I disagree with the next sentence: 

"Third column is the main comment after comparing the results of the proposed method and the references highlighted" which states how the rest methods found in the literature perform against the proposed method in the investigated method. Numerical experiments are needed and reproducible results to be provided.

Some figures are still of poor quality (a bar plot for depicting 3 average values?), while the structure of commenting the rest works into the last Section does not seem much helpful.

Author Response

(The authors gave the same response as above.)
